# Multiple Virtual Screening Strategies for the Discovery of Novel Compounds Active Against Dengue Virus: A Hit Identification Study

**Kowit Hengphasatporn** [1,2], **Arthur Garon** [3], **Peter Wolschann** [3,4], **Thierry Langer** [3], **Shigeta Yasuteru** [2], **Thao N.T. Huynh** [5], **Warinthorn Chavasiri** [5], **Thanaphon Saelee** [6], **Siwaporn Boonyasuppayakorn** [6] and **Thanyada Rungrotmongkol** [1,7,*]

[1]   Program in Bioinformatics and Computational Biology, Faculty of Science, Chulalongkorn University, Bangkok 10330, Thailand; kowith@ccs.tsukuba.ac.jp

[2]   Center for Computational Sciences, University of Tsukuba, 1-1-1 Tennodai, Tsukuba, Ibaraki 305-8577, Japan; shigeta@ccs.tsukuba.ac.jp

[3]   Department of Pharmaceutical Chemistry, Faculty of Life Sciences, University of Vienna, Althanstraße 14, 1090 Vienna, Austria; arthur.garon@univie.ac.at (A.G.); karl.peter.wolschann@univie.ac.at (P.W.); thierry.langer@univie.ac.at (T.L.)

[4]   Institute of Theoretical Chemistry, University of Vienna, 1090 Vienna, Austria

[5]   Center of Excellence in Natural Products Chemistry, Department of Chemistry, Faculty of Science, Chulalongkorn University, Bangkok 10330, Thailand; thao.huynhthanh94@gmail.com (T.N.T.H.); warinthorn.c@chula.ac.th (W.C.)

[6]   Applied Medical Virology Research Unit, Department of Microbiology, Faculty of Medicine, Chulalongkorn University, Bangkok 10330, Thailand; thanaphon.saelee@gmail.com (T.S.); siwaporn.b@chula.ac.th (S.B.)

[7]   Biocatalyst and Environmental Biotechnology Research Unit, Department of Biochemistry, Faculty of Science, Chulalongkorn University, Bangkok 10330, Thailand

*   Correspondence: thanyada.r@chula.ac.th; Tel./Fax: +66-2218-5426

**Abstract:** Dengue infection is caused by a mosquito-borne virus, particularly in children, which may even cause death. No effective prevention or therapeutic agents to cure this disease are available up to now. The dengue viral envelope (E) protein was discovered to be a promising target for inhibition in several steps of viral infection. Structure-based virtual screening has become an important technique to identify first hits in a drug screening process, as it is possible to reduce the number of compounds to be assayed, allowing to save resources. In the present study, pharmacophore models were generated using the common hits approach (CHA), starting from trajectories obtained from molecular dynamics (MD) simulations of the E protein complexed with the active inhibitor, flavanone (FN5Y). Subsequently, compounds presented in various drug databases were screened using the LigandScout 4.2 program. The obtained hits were analyzed in more detail by molecular docking, followed by extensive MD simulations of the complexes. The highest-ranked compound from this procedure was then synthesized and tested on its inhibitory efficiency by experimental assays.

**Keywords:** dengue virus; envelope protein; fragment molecular orbital; molecular dynamics; pharmacophore-based screening

## 1. Introduction

Structure-based virtual screening (SBVS) is one of the widely popular approaches to search for novel potent compounds that can interact with the target [1–3]. The challenge of SBVS is how to select the molecules to be proper inhibitors among a large number of compounds in the databases [4]. There

are several techniques that can be used for SBVS, such as pharmacophore-based virtual screening and molecular docking methods. A pharmacophore-based screening approach is one of the most advantageous tools, which matches the similarity of the three-dimensional (3D) interaction pattern or pharmacophore model of drug-like compounds with known ligands in the complexes [5–12]. Meanwhile, the molecular docking method uses a search strategy by scoring functions [13–15]. In other words, the prediction of the ligand orientation in the binding pocket can be performed and ranked by the binding interaction scores [16–20].

Pharmacophore-based screening not only generates the interaction pattern from the 3D complex structures, but also allows virtual screening to search for potent molecules from huge compound libraries. The interaction pattern defined by the hydrogen bond donor and acceptor properties, hydrophobic groups, aromatic rings of compounds that bind to a biological target, and the excluded volume is summarized as pharmacophore properties [6,11,21–24]. The important strategy of this approach is the overlays-based scoring algorithm, which is able to enhance the performance of pharmacophore-based screening [25]. The pharmacophore model can be generated from either a single protein–ligand complex or from molecular dynamics (MD) trajectories [22]. To refine the pharmacophore-based screening results, molecular docking is mostly implemented to confirm the efficiency of hits bound to the binding site and to construct the complex to determine a behavior of leads in the complex structures via all-atom molecular dynamics (MD) simulation [26,27]. As an advanced technique, MD simulation is used to observe the dynamics of all atoms in the system by mimicking the behavior of molecules in the complex structure. The combination of molecular mechanics (MM) force fields and implicit solvation model may account for binding free energy calculation. The binding free energy of focused compounds bound to the protein target is analyzed using both individual and average frames of production or interesting phases from MD simulation. The molecular mechanics generalized born surface area (MM-GBSA) and molecular mechanics Poisson–Boltzmann surface area (MM-PBSA) [28,29] methods are used to evaluate the binding efficiency of the potent compound in the complex system. The solvated interaction energy (SIE) method is a fast calculation with moderate accuracy. Based on the divide and conquer algorithm [30], the fragment molecular orbital (FMO) method can provide the detailed interaction energies with high accuracy [31,32].

Dengue fever is caused by dengue viral infection, which is annually emerged in tropical and sub-tropical regions [33,34]. Dengue virus (DENV) is a member of the Flaviviruses, consisting of four serotypes. The different DENV serotype infections can lead to severe problems by inducing the immunity of the patient or antibody-dependent enhancement (ADE) [35–37]. This infection highly impacts on the world's population, and yet there is no specific and efficient therapeutic treatment available. Although a DENV vaccine has been launched in the last few years, there are many warnings about side effects and the mortality on people who have received this vaccine [38].

DENV is a mosquito-borne enveloped virus of spherical shape, 50 nm on average. The inner core of virus is a single RNA positive-strand which is encoded into three structural proteins: Capsid (C), membrane (M), and envelope (E) protein, and seven nonstructural (NS) proteins. The enzymatic NS proteins are involved in several steps of viral infection, including NS2b/3 protease, NS3 helicase, NS5, and NS5 RNA-dependent RNA polymerase (RdRp) [39–41]. The viral genome is covered by C, M, and E protein, respectively. E protein is an important target for vaccine development against Flaviviruses, because this protein mainly interacts with the receptor of a host cell in the viral entry process [42,43]. In the classical endocytic pathway, the viral E protein is rearranged, inserted, and fused to the host endosomal membrane by low pH triggers [44–46].

Many studies have been performed to understand the inhibition mechanism of the active small molecules against DENV E protein. The kl loop or hinge region between the two domains DI and DII (K and K′ sites, Figure 1) is an important target site for antiviral agents [42,47–56]. The tip of DII (Y′ site), a conserved region among flavivirus genus such as West Nile virus, Japan encephalitis virus, Yellow fever virus, or Zika virus that immerses into the lipid bilayer of host endosomal membrane, has been suggested as the binding site for flavonoids, possibly interfering with the viral attachment

and fusion [42,57,58]. In addition, the region between DI and DIII (X' and Y' sites), a promising target for vaccine development [59,60], was also used for drug discovery [50,54]. From an in vitro and in silico study [54], a flavanone derivative (5-hydroxy-7-methoxy-6-methylflavanone, FN5Y) has been shown to interrupt the progress of the disease at the early stage of infection, targeted at E protein at the four sites (Figure 1). Herein, we aimed to search for novel compounds that can inhibit DENV at dimeric E protein using several computational tools. Flavonoids were collected from several compound databases and an in-house database based on the core structural similarity of flavonoids [61,62]. In the beginning, the pharmacophore-based screening and molecular docking methods were used for virtual screening of numerous compounds in the dataset. Moreover, the molecules needed to be evaluated regarding to their binding affinities by analyzing the behavior of complexes from MD simulation through the end-point binding free energy calculations: SIE, MM-PBSA, and MM-GBSA methods. The most preferential binding site was analyzed and characterized by the interaction energies obtained from quantum mechanical approach, FMO.

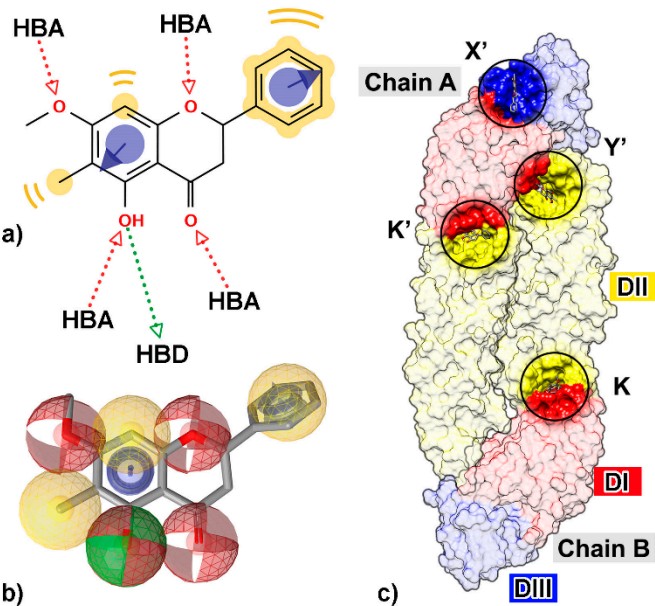

**Figure 1.** Two-dimensional (2D) structure of a flavanone derivative, 5-hydroxy-7-methoxy-6-methylflavanone (FN5Y), (**a**) with all possible chemical features such as hydrogen bond donor (HBD, green arrow), hydrogen bond acceptor (HBA, red arrow), hydrophobic property (yellow), and aromatic ring (purple circle with arrow). (**b**) The chemical features are shown as three-dimensional (3D) structure; HBD (green sphere), HBA (red sphere), hydrophobicity (yellow sphere), and aromatic ring (purple ring). (**c**) The four preferential binding sites of FN5Y on dengue virus (DENV) E protein at kl hairpin on chain B (K) and its opposite site on chain A (K'), fusion protein region which is conserved among flaviviruses (Y'), and DI/III hinge region (X'). The structure was extracted from the last snapshot at 100 ns of trajectory from the previous study [54].

## 2. Materials and Methods

The overview of this work for screening of a newly potent anti-dengue compound is summarized in Figure 2. Several screening techniques were used in this study in order to search for novel potent molecules from a large number of compounds from several databases, including the flavonoid database using MD trajectory-based virtual screening. The hit compound was then synthesized and tested experimentally.

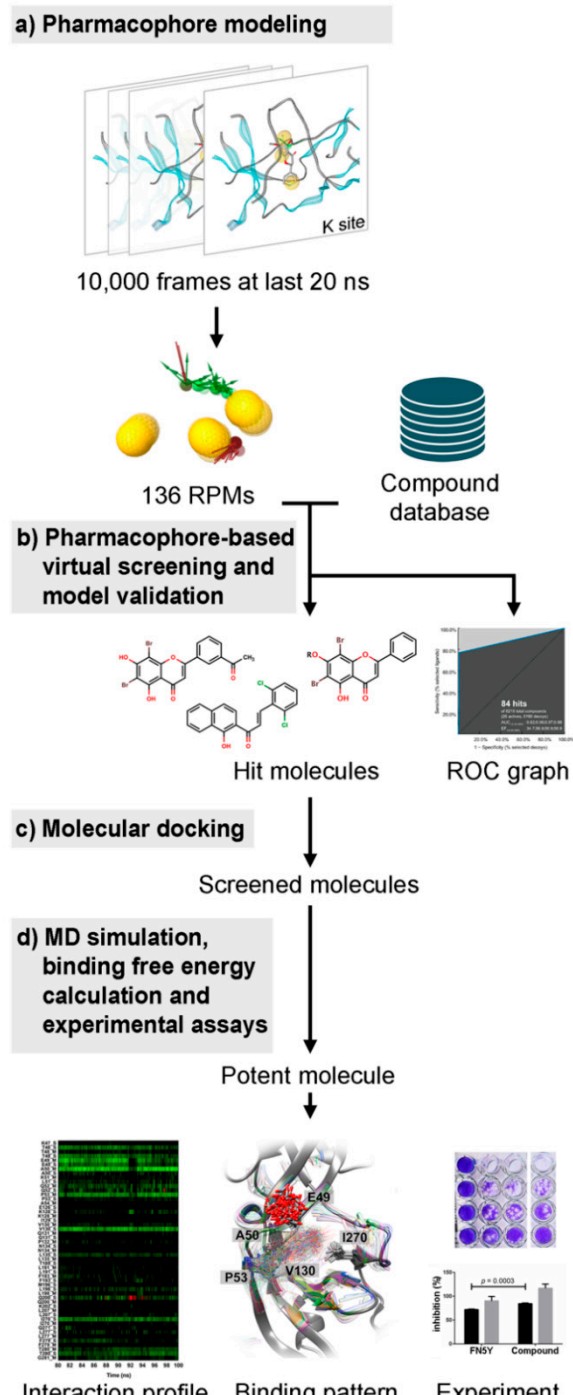

**Figure 2.** The overview of this work. (**a**) All 10,000 snapshots extracted from the last 20 ns molecular dynamics (MD) trajectories were used to generate the pharmacophore feature clustered into the representative pharmacophore model (RPM) used for screening the compound database (**b**). (**c**) The screened molecules were further investigated using molecular docking, and (**d**) subsequent MD simulations were performed. The efficacy prediction of potent flavonoid derivative was carried out by the interaction profile and binding pattern. Then, the inhibiting effect was confirmed by experimental assays.

*2.1. Pharmacophore-Based Virtual Screening*

### 2.1.1. Pharmacophore Modeling

For the preliminary screening, the intermolecular interactions of the FN5Y/E complex from our previous study [54] were used to create the pharmacophore models based on steric and electronic features. The pharmacophore feature of each of the four binding sites and FN5Y was automatically generated from MD trajectories by the LigandScout 4.2 program [6]. Note that the FN5Y/E complex prepared by docking the FN5Y molecule to E protein (1OKE.pdb) was performed by a 100 ns MD simulation in aqueous solution under periodic boundary condition (see [54] for details of simulation). A total of 10,000 frames from a trajectory over the last 20 ns simulation were used to establish pharmacophore models at individual binding site. All pharmacophore models were aligned and clustered to filter and select the unique pharmacophore model as a representative pharmacophore model (RPM) for further virtual screening.

The pharmacophore model selection was based on the common hit approach (CHA) [63]. This approach relies on the following assumption: To track a pharmacophore feature from one frame of molecular dynamics simulation to another one, the representation of this feature is not anymore based on the type of pharmacophore interaction (hydrophobic, aromatic, H-bond donor or acceptor, negative or positive ionizable area) and its 3D coordinates, but on its type and the atoms involved for its creation (from both protein and ligand side). In this way, if a feature involves the same partners, even with a small difference of position, it can be linked from different frames. On the other hand, if the position shift is too important, then the partners involved are different than before and the feature is therefore not the same. This representation allows detecting all the unique features observed in the simulation; in this case, for 1536 unique combinations of type and involved partners. Now that the features are tracked, we can consider the pharmacophore models as the sets of features. Pharmacophores are clustered by composition, keeping only the unique sets (pharmacophores) observed, here 136 RPMs. Finally, as most of the unique observed pharmacophores were present in multiple frames, the first occurrence of each of these models was selected to be the representative pharmacophore model of their cluster. This is an arbitrary selection, but we assume that all the models are similar, as they involve the same atoms from both the protein and ligand.

### 2.1.2. Database Construction

In this study, we focused on the flavonoids reported in the previous study [54]. The first compound library was constructed by the similarity search method. Six subclasses of flavonoid structures were used as templates to search for similar molecules from the Zinc database by 80% Tanimoto-based similarity [64,65]. The second compound dataset was downloaded from the TimTec compound library [66]. In total, the compound library consisted of 996 compounds; 450 molecules from Zinc, 507 molecules from TimTec, and 39 in-house designed molecules. For each molecule, conformers up to 400 conformations were generated with the "icon best" feature. Therefore, a total of 398,400 ligand conformations were saved as the local database for further steps.

### 2.1.3. Virtual Screening

Virtual screening was performed by the pharmacophore-based screening technique [63]. All compounds in library were screened for each RPM by using "get best matching conformation" in retrieval mode, and checked exclusion volume based on pharmacophore-fit scoring function (Equations (1) and (2)). The hits from each screening iteration were collected and refined by the common hits approach (CHA), developed by merging and rescoring each RPM run to generate the single hit-list of screened ligands. The steps between pharmacophore generation and the final hit-list were implemented

by the KNIME workflow program [67]. The final hit-lists were created in SDF file format and visualized by the LigandScout 4.2 program [6].

$$S_{RMS} = 9 - (3 \times \min(\text{RMS}_{FP}, 3)) \tag{1}$$

$$S_{FCR} = c \times N_{MFP} + S_{RMS} \tag{2}$$

In Equations (1) and (2), $S_{RMS}$ is the feature count/RMS distance score, $RMS_{FP}$ is the root mean squared (RMS) of the matched feature pair distance, $S_{FCR}$ is the matched feature pair RMS score, $c$ stands for the weighting factor for the number of matched feature pairs, and $N_{MFP}$ is the number of geometrically matched feature pairs [6].

### 2.2. Model Validation

The robustness of screening results was diagnosed and validated by receiver operating characteristic curve and the area under this curve (ROC-AUC). This curve illustrates the performance of pharmacophore screening by comparing the results between active ligands and decoys in the datasets. The area under ROC curve (AUC) was used for method validation [68]. For comparison, a set of inactive compounds (decoys) designed by mimic from the active molecule structure was obtained from the Zinc database [69]. The ROC values were gathered from the individual RPM screening with active and decoy datasets. Then, the values were plotted, analyzed, and interpreted in terms of sensitivity (true positive rate, TPR) and specificity (false positive rate, FPR) of screening results. Sensitivity (Equation (3)) and specificity (Equation (4)) measurements were used to validate the pharmacophore-based screening results.

$$Sensitivity = \frac{Selected\ active\ ligands}{All\ selected\ ligands} \tag{3}$$

$$Specificity = \frac{Dicarded\ inactive\ ligands}{All\ inactive\ ligands} \tag{4}$$

### 2.3. Molecular Docking

The 26 screened compounds were then independently docked into the four binding sites (K, K′, X′, and Y′; Figure 1c) on dimeric E protein with 100 runs based on two different algorithms, simulated annealing algorithm [70] in CDOCKER software of Discovery Studio 2.5 package and Genetic algorithm (GA) in iGEMDOCK 2.1 [71]. The protein structure for molecular docking was extracted from the last snapshot of FN5Y/E simulation from the previous study [54], without ligands, ions, and water molecules. The protonation state of screened compounds was determined at neutral pH by ChemAxon software [72,73]. The geometry of ligands was optimized by HF/6-31G(d) level of theory using Gaussian 09 (Gaussian Inc., Wallingford, CT, USA). More details of molecular docking can be found in the recent studies [54,56,58]. The docked conformation with the best interaction energy and fitness score at each specific binding site was selected as an initial complex structure for all-atom MD simulations.

### 2.4. MD Simulations and Binding Energy Calculations

The top three docked complexes ranked from CDOCKER interaction energies and iGEMDOCK fitness scores were used for a 100 ns MD simulation with periodic boundary condition using the AMBER 16 program [74]. The partial charges of each ligand were prepared in accordance with the standard protocol [75–77], while the other parameters were taken from the general AMBER force field [74]. The AMBER ff14SB force field [78] was applied on the E protein. All system preparation, minimization, and MD simulation at 300 K were set as in our previous studies on FN5Y and cardol triene in complex with DENV E protein [54,56]. The system stability of each complex was accessed by root mean square displacement (RMSD), as plotted in Figures S1 and S2 (in Supplementary Materials). The solvated interaction energy (SIE) method [28,79] was applied on the 100 snapshots taken from the

last 40 ns in order to determine the binding free energy of the complex. Among the three simulated complexes, the simulation of the best complex with lowest SIE binding free energy at all four sites was extended to 500 ns.

To identify the key binding residue at each site, per-residue decomposition free energy calculations with the MM-GBSA method [79] were applied on the 500 ns trajectories from the last 100 ns. A more detailed ligand binding at the most preferential binding site was refined using FMO at the second-order Møller–Plesset (MP2) [80] level with the 6-31G(d) basis set using GAMESS [81,82]. Only the residues within 8 Å sphere of ligand from the last snapshot were calculated by a separation into small fragments between Cα–C atoms, i.e., one residue per fragment [83]. Paired interaction energy (PIE and PIEDA, $\Delta E_{ij}^{int}$) is a summation of electrostatic ($E_{ij}^{ES}$), charge exchange ($E_{ij}^{EX}$), charge transfer ($E_{ij}^{CT+mix}$), and dispersion ($E_{ij}^{DI}$) [84]:

$$\Delta E_{ij}^{int} = \Delta E_{ij}^{ES} + \Delta E_{ij}^{EX} + \Delta E_{ij}^{CT+mix} + \Delta E_{ij}^{DI} \qquad (5)$$

### 2.5. Synthesis of 6,8–Dibromo-5-Hydroxy-7-Dodecyloxyflavone

The potent molecule, 6,8-dibromo-5-hydroxy-7-dodecyloxyflavone or F18, was synthesized in two steps, as shown in Figure 3. Firstly, a solution of chrysin (2 mmol) in dry acetone was treated with anhydrous $K_2CO_3$ and dodecyl bromide (2 mmol). The mixture was refluxed under $N_2$ atmosphere overnight, cooled to room temperature, and filtered. The solid $K_2CO_3$ was washed with acetone. Evaporation of the combined organic solvents under reduced pressure furnished a residue, which was purified by silica gel column, leading to 5-hydroxy-7-dodecyloxyflavone as a pale yellow solid (56%). [1]H NMR (400 MHz, CDCl$_3$) δ$_H$ 7.89 (m, 2H), 7.53 (m, 3H), 6.67 (s, 1H), 6.50 (d, J = 2.2 Hz, 1H), 6.37 (d, J = 2.2 Hz, 1H), 4.03 (t, J = 6.5 Hz, 2H), 1.89–1.25 (m, 20H), 0.87 (m, 3H).

Secondly, 5-hydroxy-7-dodecyloxyflavone (0.10 mmol) was dissolved in dichloromethane (10 mL) before adding N-bromosuccinimide (0.10 mmol) at room temperature within 5 min. The mixture was stirred at room temperature for 6 h. After the reaction completed (monitoring by TLC), the mixture was evaporated under reduced pressure. The crude product was purified by silica gel column (hexane/dichlromethane/acetone 7.5:2.5:0.1) to afford the desired compound (25% yield) as a yellow powder. [1]H NMR (400MHz, CDCl$_3$): 8.03–8.01 (m, 2H), 7.61–7.55 (m, 3H), 6.82 (s, 1H), 4.14 (t, J = 7.0 Hz, 2H), 1.96–1.91 (m, 2H), 1.29 (b, 18H), 0.88 (t, J = 7.0 Hz, 3H). The NMR spectra are given in Figures S5 and S6 (Supplementary Materials).

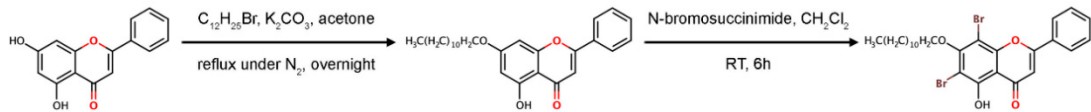

**Figure 3.** Synthesis of 6,8-ibromo-5-hydroxy-7-dodecyloxyflavone, F18.

### 2.6. Experimental Assays

Vero (ATCC®CCL-81) and LLC/MK2 (ATCC®CCL-7) cells were maintained as previously described [85]. A DENV2 (New Guinea C strain) reference strain was propagated in Vero cell line with Medium-199 supplemented with 1% fetal bovine serum, 100 I.U./mL penicillin, 100 μg/mL streptomycin, 10 mM HEPES, and 10% NaHCO$_3$ at 37 °C under 5% CO$_2$.

First, to prove that the addition of DMSO and acetone to 1% did not affect the cell viability, LLC/MK2 ($10^4$) cells were seeded into each well of 96-well plates and incubated overnight at 37 °C with 5% CO$_2$. Cells were then replaced with maintenance media (minimal essential media (MEM) supplemented with 1% fetal bovine serum, 100 I.U./mL penicillin, 100 μg/mL streptomycin, 10 mM HEPES, and 10% NaHCO$_3$) alone, or containing 1% acetone or 1% DMSO. Moreover, F18 was dissolved in acetone and added to the final concentration of 10 μM in maintenance media containing 1% acetone. Cells were incubated for 48 h before analysis by spectrophotometer (A$_{450nm}$) (VICTORTM X3,

PerkinElmer, MA, USA). Each condition was done in triplicate. Percent cell viability in each condition was calculated using maintenance media alone as a 100% infectivity.

Next, to screen the DENV2 inhibition of F18, LLC/MK2 ($5 \times 10^4$) cells were seeded into each well of 24-well plates and incubated overnight at 37 °C with 5% $CO_2$. The cells were then infected with DENV2 (M.O.I. of 0.1) in maintenance media and incubated at 37 °C for 1 h with gentle rocking every 15 min. The compound F18 was dissolved in acetone and added to the final concentration of 10 μM in 1% acetone in maintenance media. The maintenance media with no virus addition were used as a negative control (neg). The virus-addition in maintenance media containing 1% acetone (pos A) or DMSO (pos D) were used as positive controls. Cells were incubated at 37 °C for 3 days and supernatants were collected for plaque titration [86]. Plaques were counted to plaque forming unit (pfu/mL) [87]. The experiment was done in triplicate. Percent compound inhibition was calculated using maintenance media alone as a 100% infectivity.

To study the effective concentration ($EC_{50}$) against DENV2 infectivity, LLC-MK2 ($5 \times 10^4$) cells were seeded into each well of 24-well plates and incubated at 37 °C for 1 h, with gentle rocking every 15 min. The F18 were serially diluted in acetone and added to the experimental to the final concentrations of 0, 0.5, 1, 1.5, 2, 2.5, 5, 7.5, 10, 25, and 50 μM. Cells were incubated at 37 °C for 3 days and supernatants were collected for plaque titration [86]. Effective concentration of the compound was calculated using non-linear regression analysis.

## 3. Results and Discussion

### 3.1. MD Pharmacophore-Based Virtual Screening

A total of 10,000 MD snapshots extracted from production phase of the DENV2 FN5Y/E complex [54] were used to generate pharmacophore features using the LigandScout 4.2 program (Figure 4) [6]. All RPMs were aligned and merged, then they were used as the template for pharmacophore-based virtual screening [63]. According to Figure 4, the major pharmacophore feature among the four binding regions are mainly described by hydrophobic properties of the aromatic ring system of the FN5Y molecule (yellow sphere). Additionally, the 3D interaction pattern reveals hydrogen bond donor (green arrow), hydrogen bond acceptor (red arrow), and an excluded volume (grey sphere). These properties are collected and aligned into 40,000 RPMs and filtered into 136 RPMs in order to reduce the computational afford. All of these RPMs were obtained from each binding site: K (31), K′ (26), X′ (42), and Y′ (3), with 1536 pharmacophore features. The pharmacophore-based virtual screening method was performed on each RPM using Hungarian matching algorithm implemented in the LigandScout 4.2 program [6]. The screening results from 136 different RPMs were merged and rescored. The consensus molecules were ranked by the common-hits approach (CHA) [63]. From the 996 compounds in the focused library, the result shows that only 26 flavonoids have a higher CHA score than 50 (Figure 5, right): From in-house (F) 17 molecules, including FN5Y, TimTec (ST) 7 molecules, and Zinc (ZINC) 1 molecule (chemical structures shown in supplemental Figure S3). F15 shows the highest CHA score of 77, whereas ZINC000236568961 shows the lowest CHA score of 50. From up to 400 generated conformations of F15, its 57 conformers were well fitted with the 82 RPMs of FN5Y/E complex (60% from 136 RPMs).

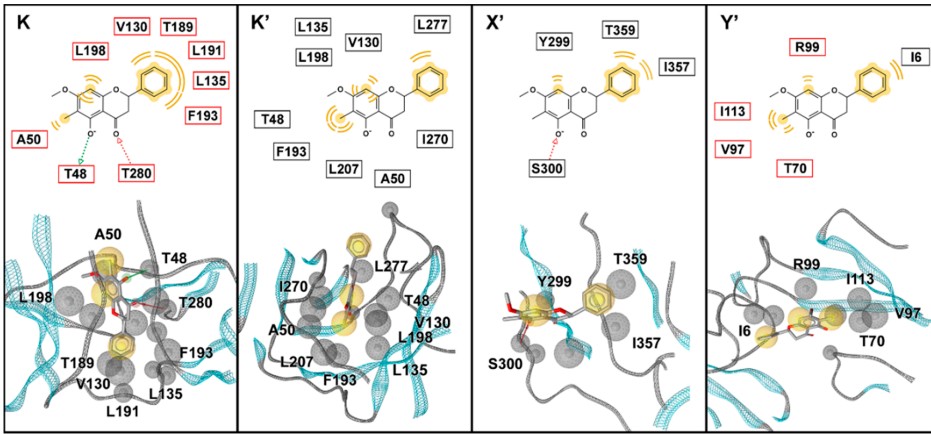

**Figure 4.** The 2D and 3D pharmacophore models of the FN5Y/E complex and the interacting residues in each pocket (K, K', X', and Y') extracted from the first snapshot of the last 20 ns MD trajectories [54]. The pharmacophore features are represented as green arrows (HBD), red arrows (HBA), yellow spheres (hydrophobic property), and grey spheres (excluded volume). The interacting residues from chains A and B are labeled in black and red boxes, respectively.

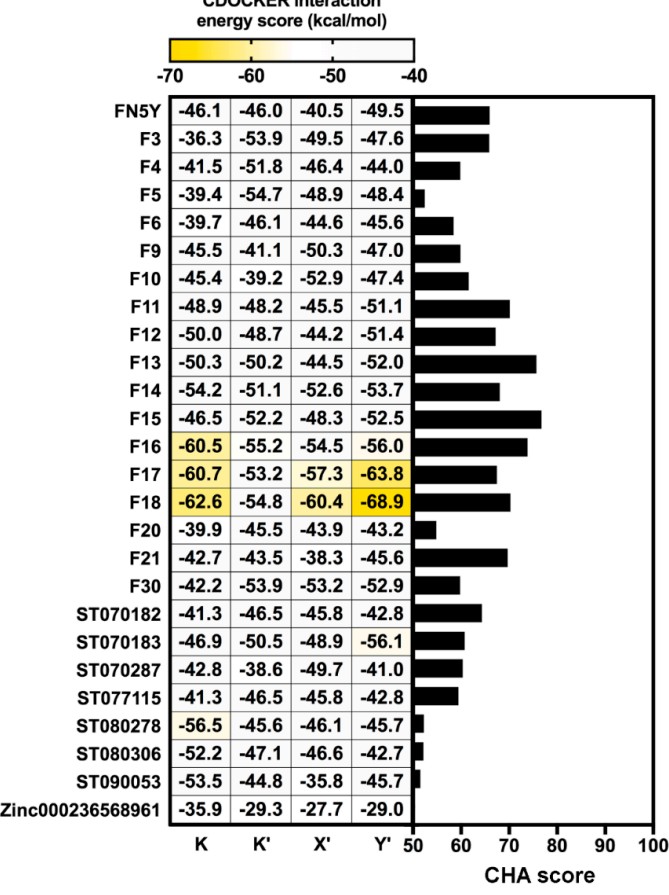

**Figure 5.** Comparison of the common hit approach (CHA) scores of screened molecules bound to all binding sites (**right**) and CDOCKER interaction energy scores (**left**). The screened molecules with CHA scores over 50 are plotted as a bar graph. All of these molecules were docked into four binding sites of DENV E protein using CDOCKER software.

*3.2. Hit Compounds*

A total of 26 screened molecules, including FN5Y, were docked into the four specific binding sites (K, K′, X′, and Y′) of DENV2 E protein. The results show all of the flavonoid derivatives from three databases can fit well into every binding pocket based on CDOCKER interaction energy score (Figure 5, left), which is similar to the pattern of iGEMDOCK's fitness score (Figure S4). Particularly, in-house flavone derivatives F16, F17, and F18 are ranked in the top three with the lowest CDOCKER interaction energy score (from −53.24 to −68.93 kcal/mol). These molecules might be potent ligands for designing broad-spectrum inhibitors against DENV at E protein target. The common structure of the top three candidate molecules contains long hydrophobic tails –$(CH_2)_8$, –$(CH_2)_{10}$, and –$(CH_2)_{12}$ (Figure S3), interacting mostly with the hydrophobic and polar uncharged residues of E protein (Table 1). All hit compounds shared the binding residues with the known active compound FN5Y as follows: Hydrophobic (I6, V130, and F193), polar uncharged (T155 and T359), and positively charged (K247 and K295) amino acids. In order to characterize the behavior of ligand in the complexes, the conformations of F16, F17, and F18 at individual binding sites (K, K′, X′, and Y′) were then studied by MD simulations.

**Table 1.** Contributed residues of hit molecules bound at the four binding sites on the DENV dimeric E protein in comparison to FN5Y molecule. The binding residues of all hit compounds shared with the known active compound FN5Y are shown in bold text. The residue contributions are shaded according to the CDOCKER interaction energy scores in Figure 5.

| | K | K′ | X′ | Y′ |
|---|---|---|---|---|
| FN5Y | T48<br>A50<br>V130<br>L135<br>L191<br>F193<br>L198<br>L207<br>L277 | A50<br>K128<br>L135<br>F193<br>L198<br>I270 | K295<br>Y299<br>I357<br>T359 | I6<br>T155<br>V97<br>K247 |
| F16 | **T48**<br>E49<br>**A50**<br>**V130**<br>**F193**<br>**L198**<br>**L207**<br>I270<br>**L277** | T48<br>E126<br>**K128**<br>**L135**<br>**F193**<br>**I270**<br>L277 | **K295**<br>**Y299**<br>**I357**<br>**T359** | **I6**<br>**T155**<br>T70<br>**K247** |
| F17 | T48<br>E49<br>**A50**<br>**V130**<br>**L135**<br>T189<br>**L198** | K47<br>**A50**<br>V130<br>**L135**<br>**F193**<br>**L198** | **K295**<br>**Y299**<br>**I357**<br>**T359** | **I6**<br>**T155**<br>T69<br>T70<br>I113<br>**K247** |
| F18 | **V130**<br>**L191**<br>F193<br>Q200<br>**L207**<br>**L277** | K47<br>**A50**<br>V130<br>**F193**<br>**L198** | **K295**<br>T303<br>**T359** | **I6**<br>**T155**<br>T69<br>T70<br>**K247** |

### 3.3. Virtual Screening Validation

The performance of pharmacophore-based screening results and the ability of identification between 6190 decoys and 26 active compounds were achieved by the receiver operating characteristic (ROC) plot. Note that the area under the curve (AUC) represents the quality of ROC plot. The AUC value of >0.50 suggests that the results from this method are reliable, in which the active molecules are likely screened [63]. In Figure 6, the AUC values are 0.82 (1%), 0.96 (5%), 0.97 (10%), and 0.88 (100%); therefore, the hit compounds from this pharmacophore-based screening are acceptable for further antiviral drug development.

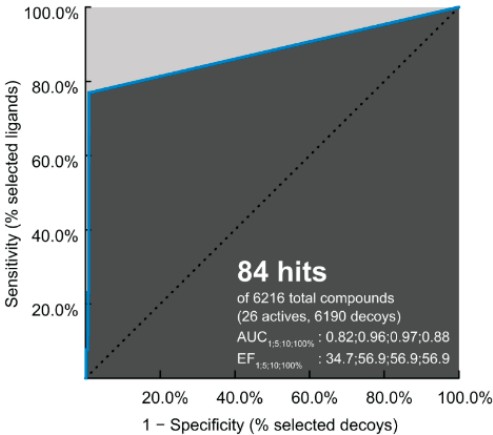

**Figure 6.** Receiver operating characteristic (ROC) plot of pharmacophore model applied to DENV E protein. The enrichment factor (EF) and area under the curve (AUC) are given for 1, 5, 10, and 100% of the database.

### 3.4. Potent Compounds

To identify the most potent flavone derivative binding to DENV E protein, the binding free energy calculation based on the SIE method was applied on the three complexes (F16, F17, and F18) using the last 40 ns trajectories in comparison to the known compound, FN5Y. The SIE binding free energy in solvation ($\Delta G_{bind}$) was estimated by a summation of non-polar components ($\Delta E_{vdW}$) and Coulomb interactions ($\Delta E_{elec}$), as well as desolvation free energy contributions [28,79]. The $\Delta G_{bind}$ values of the considered complexes are plotted in Figure 7, while the energy components are given in Table S1 (Supplementary Materials).

In Figure 7, the calculated $\Delta G_{bind}$ values of F16, F17, and F18 complexes are in a range of −10.48 to −13.06, −8.56 to −10.81, and −11.11 to −14.42 kcal/mol, respectively. These predictions suggest the binding affinity of these halogenated flavones with the E protein are significantly higher than that of FN5Y (−3.43 to −4.35 kcal/mol). In addition, vdW interaction is the major contribution for ligand/E protein complexes ($\Delta E_{vdW}$; Table S1). F18 has the highest efficacy to bind on the surface of the E protein at all sites, and its preferential binding site is located at the kl loop region, K and K′. The bromine substitutions in F18 could strengthen the interactions at all pockets [85,88–90], which may lead to an improved drugability of this potent molecule [91].

Simulation on the F18/E complex was then extended to 500 ns to investigate the binding pattern and interaction profile of F18. Per-residue decomposition free energies ($\Delta G_{residue}$) for F18 at the four different binding sites over and along the last 100 ns simulation are given in Figure 8a,b, respectively. The F18 orientations inside each binding site are depicted in Figure 9, where the surrounding residues with $\Delta G_{residue} \leq -1$ kcal/mol are labeled. There are several residues important for F18 binding at K (E49, A50, P53, K128, L135, L198, and Q200), K′ (T48, E49, A50, L135, L198, I270, Q271, and T280), X′ (H149, K157, H158, and Y299), and Y′ (V97, I113, P243, K247, and Q248). By considering the contributing residues and their stabilization energies, again, F18 likely prefers to interact at the K and K′ binding

regions. Then, the FMO calculation was used to rigorously reveal the strong paired interaction energy ($E_{ij}$) [92] between F18 and the residues at the K site.

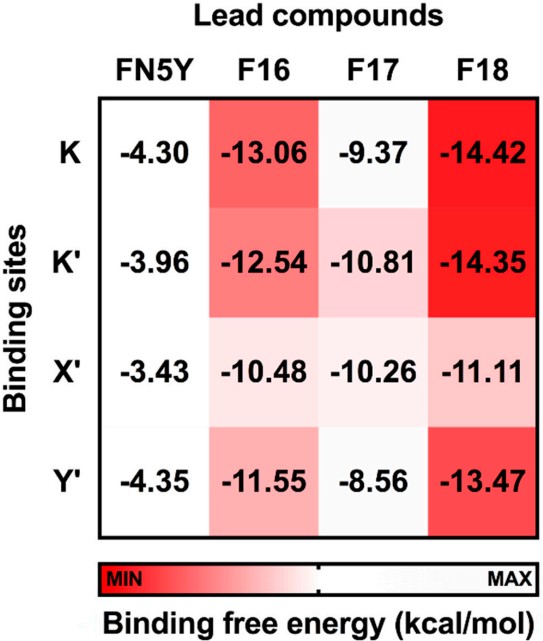

**Figure 7.** Solvated interaction energy (SIE) binding free energy ($\Delta G_{bind}$) in kcal/mol of FN5Y, F16, F17, and F18 binding at the four different sites on DENV dimeric E protein, K, K′, X′, and Y′.

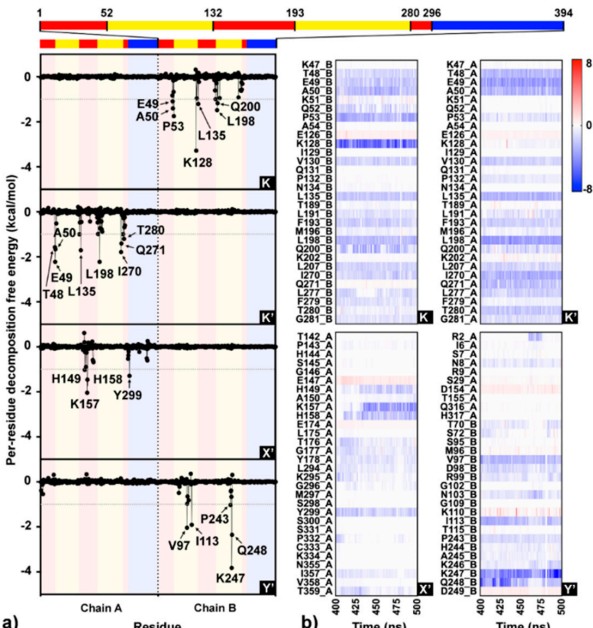

**Figure 8.** (**a**) Interaction profile of F18 binding to the four sites (K, K′, X′, and Y′) on dimeric DENV E protein over the last 100 ns MD trajectories based on molecular mechanics generalized born surface area (MM-GBSA) per-residue decomposition free energy calculation. Each monomeric E protein contains three different domains: DI (red), DII (yellow), and DIII (blue). (**b**) Per-residue decomposition free energies for the residues within 8 Å sphere of F18 per time are shown as heat map shaded by red-to-blue spectrum.

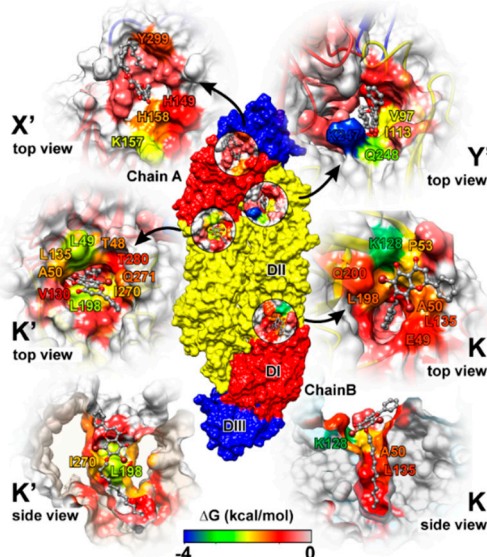

**Figure 9.** F18 binding orientations at the four binding sites, where the interacting residues are colored according to the energy values from Figure 9a. The residues with decomposition free energy contribution lower than −1 kcal/mol are labeled.

At the K site, the residue K128 shows the highest stabilization ($\Delta G_{residue}$ of −3.24 kcal/mol in Figure 8a), mainly through electrostatic attraction with the negatively charged oxygen and partial dispersion to bromine [93] on F18 flavone (−85.24 and −3.99 kcal/mol from FMO MP2/6-31G(d) calculation; Figure 10). Although the two residues Q200, a key amino acid for cyanohydrazone (3-110-22) binding [55], and E49 favorably stabilize F18 (Figure 8a,b), they show electrostatic repulsion (34.03 and 31.37 kcal/mol; Figure 10) with the two bromine atoms (Figure 9). The other residues, A50, P53, L135, and L198, contribute to stabilize the non-polar tail of F18.

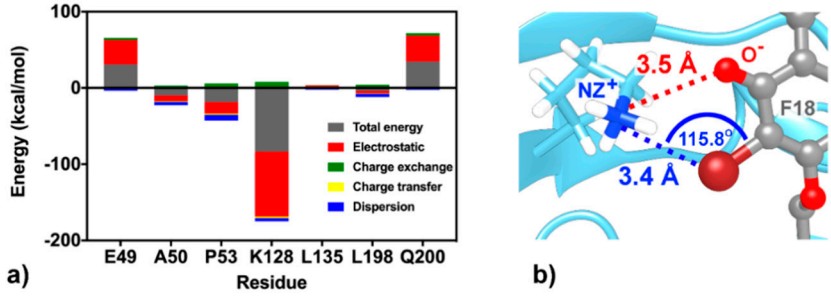

**Figure 10.** (**a**) FMO2-MP2/6-31G(d) binding energy components between the K-site residues and F18. (**b**) Electrostatic attraction (red dashed line) and dispersion (blue dashed line) between K128 and flavone moieties.

### 3.5. Efficacy of Potent Compound Against DENV

Ten micromolar (μM) of compound F18 was experimentally tested for DENV2 inhibition and toxicity in cell-based system. Note that the compound was completely dissolved in acetone, but not in dimethylsulfoxide (DMSO), which was the primary solvent in cell-based drug screening. Therefore, the cell viability under different solvents was first analyzed (Figure 11a). Results show that LLC/MK2 cells are similarly viable in maintenance media alone or containing 1% acetone or 1% DMSO. In addition, the cell viability in 10 μM F18 was insignificantly higher than those in maintenance media with 1% acetone ($p = 0.0856$). Next, the compound screening for DENV2 inhibition showed about a 1-log reduction of the plaque titer. In other words, 10 μM F18 inhibited DENV2 infectivity to 90% in a cell-based system (Figure 11b). This experiment was performed in triplicate in two independent experiments.

Moreover, the F18 efficacy was further analyzed (Figure 11c) with three technical replicates in an independent experiment, and results were compared with those of FN5Y previously reported with three technical replicates in three independent experiments at the same condition (Figure 2A of [54]). Our results indicate that the F18 expressed DENV2 inhibition 3.83 times over FN5Y with an $EC_{50}$ of 4.17 μM and 15.99 μM, respectively. We conclude that the results from cell-based assay support the computational calculation.

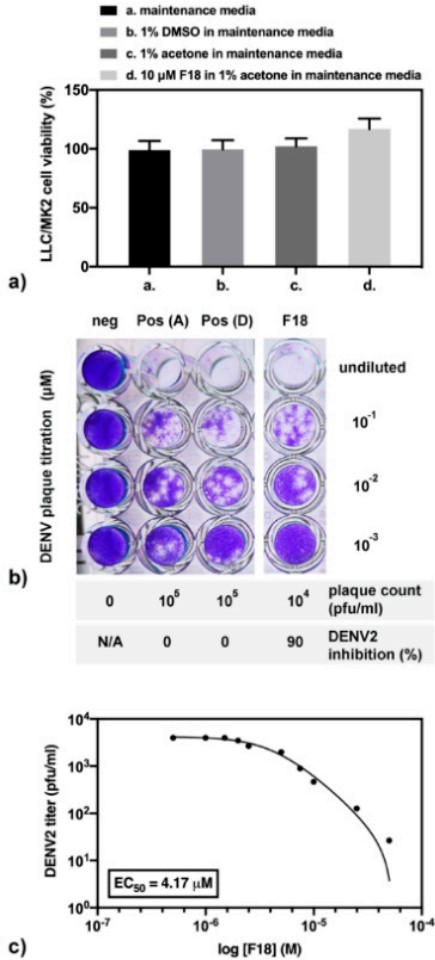

**Figure 11.** The cell-based assay of F18 in (**a**) cell viability; (**b**) compound screening for inhibition of DENV2 infectivity; and (**c**) efficacy study.

## 4. Conclusions

MD pharmacophore-based virtual screening was applied in the present work to screen the hit compounds targeting DENV E protein using the 3D Hungarian matching and overlays-based scoring algorithm. The 40,000 pharmacophore models, generated from the known potent compound FN5Y in complex with E protein, were filtered into 136 RPMs for screening a large number of compounds from the Zinc (ZINC), TimTec (ST), and in-house (F) databases. Based on a CHA score >50, the 26 hit compounds (17 F, 7 ST, and 1 ZINC) were selected. The molecular docking results from CDOCKER and iGEMDOCK suggested the three best fitting compounds, halogenated flavones F16, F17, and F18, toward the dimeric E protein. Consequently, 100 ns MD simulations and SIE binding free energy calculations were used to find the most potent compound (F18 as a result). The complex was then simulated until 500 ns. The flavone F18 can interact strongly at the four sites K, K', X', and Y' on E protein surface, particularly the kl loop region, K and K'. From MM/GBSA per-residue decomposition free energy and FMO MP2/6-31G(d) calculations, the residue K128 plays a key role in the stabilization

of F18 binding at the K site via electrostatic attraction with negatively charged oxygen and partial dispersion to bromine, while the long tail hydrocarbon was feasibly stabilized by the non-polar residues, e.g., A50, L135, and L198. Thus, F18 was then synthesized and tested for its DENV inhibition efficiency by plaque and cell-based assays, and the results showed that F18 was more potent than the previously reported FN5Y. Altogether, the obtained data suggest this compound as a novel antiviral drug candidate against dengue virus.

**Supplementary Materials:** Supplementary materials can be found at http://www.mdpi.com/2218-0532/88/1/2/s1. Root mean square displacement (RMSD) plot for the protein backbone and all ligand atoms of flavones/E protein complexes; RMSD plot for the protein backbone and all ligand atoms of F18/E protein complexes; Two-dimensional (2D) structures of the 26 hit compounds; CDOCKER interaction energies (black) and iGEMDOCK's fitness scores (grey) of the 26 hit compounds docking to dengue virus (DENV) E protein; Interaction components of screened molecules docking to each binding regions; Solvated interaction energy (SIE) Binding free energy (Δ) in kcal/mol for each flavone bound to DENV E protein at the 4 sites; NMR spectra of compound 18.

**Author Contributions:** Conceptualization, T.R., P.W. and K.H.; methodology, W.C., T.N.Y.H., S.B., T.S., A.G. and K.H.; software, T.R., S.Y., T.L. and P.W.; validation, P.W., A.G. and K.H.; formal analysis, K.H. and A.G.; investigation, K.H., S.B. and W.C.; resources, T.R., S.Y., T.L. and W.C.; data curation, K.H.; writing—original draft preparation, K.H.; writing—review and editing, P.W. and T.R.; visualization, K.H.; supervision, T.R.; project administration, T.R.; funding acquisition, T.R., S.B. and P.W. All authors have read and agreed to the published version of the manuscript.

**Funding:** This research was financially supported by the National Research Council of Thailand and Health Systems Research Institute (HSRI). Through travel grants for a short research visit, this research was also supported by the ASEAN-European Academic University Network (ASEA-UNINET).

**Acknowledgments:** K.W. thanks the Center for Computational Sciences, University of Tsukuba, for supporting a research visit. The Center of Excellence in Computational Chemistry (CECC) and the Vienna Scientific Cluster (VSC-2) are acknowledged for facilities and computing resources. The authors also thank the Research Chair Grant, the National Science and Technology Development Agency (NSTDA), Thailand.

**Conflicts of Interest:** The authors declare no conflict of interest.

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
