# Peer review of "Multiple Virtual Screening Strategies for the Discovery of Novel Compounds Active Against Dengue Virus: A Hit Identification Study"

_scipharm, doi:10.3390/scipharm88010002_

Round 1
Reviewer 1 Report
Very nice manuscript I recommend publication of the findings. Nevertheless, the authors should check the spelling in the text, there are many mistakes. Check and revise references as well.
Author Response
Point 1: Very nice manuscript I recommend publication of the findings. Nevertheless, the authors should check the spelling in the text, there are many mistakes. Check and revise references as well.
Response 1: Thank-you for your careful review. All of these suggestions above has been corrected in the revised manuscript.
Reviewer 2 Report
The presented manuscript contains a very interesting screening workflow, followed by an experimental validation of the retrieved hits. The work was well conducted and I suggest to publish the paper after minor revision:
the of the characterization of synthesized compounds should be improved inserting as supplementary the spectra of 1H and 13C (that the authors should add also in the main text) the number of the independent in vitro experiments for determining the efficacy of the compounds should be reported. the authors should remove the section at the end of materials and methods since it is the part of the template.
Author Response
Point 1: The of the characterization of synthesized compounds should be improved inserting as supplementary the spectra of 1H and 13C (that the authors should add also in the main text).
Response 1: First of all, we would like to thanks for your beneficial comments and suggestions that allow us to improve the quality of this manuscript. The NMR spectroscopy of F18 compound was added as Fig. S5-6 on page S7 in Supplementary Materials. And we are also added the detail of NMR spectroscopy in line 235-236.
Point2: The number of the independent in vitro experiments for determining the efficacy of the compounds should be reported.
Response 2: The technical and biological replicates were added as suggested in the current manuscript (line 418-421).
Point3: The authors should remove the section at the end of materials and methods since it is the part of the template.
Response 3: Thank you for pointing it out. It has been removed from this section.
Reviewer 3 Report
The presented manuscript contains comprehensive data started from computational studies and hit selection followed by in vitro activity assays. The work is well design and performed. In my opinion the work could be published in present form preceded by spellcheck.
Author Response
Point 1: The presented manuscript contains comprehensive data started from computational studies and hit selection followed by in vitro activity assays. The work is well design and performed. In my opinion the work could be published in present form preceded by spellcheck.
Response 1: We would like to thank the first reviewer by his/her positive comments and suggestions.